# Evaluation of the Containment of Antimicrobial-Resistant *Salmonella* Species from a Hazard Analysis and Critical Control Point (HACCP) and a Non-HACCP Pig Slaughterhouses in Northeast Thailand

**DOI:** 10.3390/pathogens9010020

**Published:** 2019-12-24

**Authors:** Xin Wu, Fanan Suksawat, Allen L. Richards, Seangphed Phommachanh, Dusadee Phongaran, Sunpetch Angkititrakul

**Affiliations:** 1Research Group for Animal Health Technology, Faculty of Veterinary Medicine, Khon Kaen University, Khon Kaen 40002, Thailand; wuxinddl@hotmail.com (X.W.); sjirap@kku.ac.th (F.S.);; 2Research and Diagnostic Center for Emerging Infectious Diseases, Khon Kaen University, Khon Kaen 40002, Thailand; 3Uniformed Services University of the Health Sciences, Bethesda, MD 20814, USA; 4Faculty of Agriculture, National University of Laos, Vientiane 7322, Laos

**Keywords:** antimicrobial resistance, *Salmonella* spp., Hazard Analysis and Critical Control Point, pig slaughterhouse, slaughtering contamination rate

## Abstract

To evaluate the containment of antimicrobial-resistant (AMR) *Salmonella* contaminations of a HACCP slaughterhouse (HACCP SH) and a non-HACCP slaughterhouse (non-HACCPSH), 360 paired pig rectal (representing the farm pig status) and carcass samples (representing the contamination) were collected equally from the two slaughterhouses that serviced 6 and 12 farms, respectively, in Northeast Thailand (n = 720). The purified *Salmonella* isolates were serotype identified, antimicrobial susceptibility tested, and pulsed-field gel electrophoresis (PFGE) assessed. Four evaluations of two slaughterhouses were examined: (1) the means of slaughtering contamination rates (SCR) (to evaluate the contamination level by averaged farm SCRs): the HACCP SH decreased contamination (SCR: −48.89% ± 8.80%, n = 6), whereas the non-HACCP SH increased (SCR: 14.31% ± 9.35%, n = 12). (2) The serotype diversity: the HACCP SH decreased the diversity from the rectal group (110 isolates, 9 serotypes) to carcass group (23 isolates, 3 serotypes), whereas there was no decrease in the non-HACCP SH (rectal group (66 isolates, 14 serotypes) and carcass group (31 isolates, 10 serotypes)). (3) The AMR patterns: the HACCP SH decreased from rectal group (96 isolates, 7 patterns) to carcass group (22 isolates, 1 pattern), whereas there was no decrease from the non-HACCP SH rectal group (22 isolates, 7 patterns) to carcass group (48 isolates, 8 patterns). (4) The estimated indirect contamination rate (by serotype screening and PFGE confirmation): the HACCP SH was 60.87% (14/23), whereas the non-HACCP SH was 98.48% (65/66). This study indicates that both the slaughterhouses keep a high level of indirect contamination; the HACCP SH decreases *Salmonella* contaminations and reduces the AMR patterns, the non-HACCP SH increases contaminations.

## 1. Introduction

*Salmonella* spp. are dangerous pathogens often found in the food chain. *Salmonella* (non-typhoidal) are one of the four chief global causes of 550 million annual diarrheal disease cases, announced by the World Health Organization [1]. *Salmonella enteritidis* was the most common detected agent of the total 5079 reported food-borne outbreaks in Europe in 2017 [2]. Antimicrobial resistant (AMR, Greenwood village, CO, USA) *Salmonella* makes salmonellosis more dangerous, by affecting the success of antimicrobial therapy [3] and by spreading antimicrobial resistance to other agents [4]. Moreover, it has been reported that the high prevalence of antimicrobial-resistant (AMR; Greenwood village, CO, USA) *Salmonella* spp. from food animals has been transferred to humans [3]. In particular, the pig industry provides the most meat consumed by people in the world and it is well known that pigs shed and their pork products provide a large amount of *Salmonella* spp. [5].

The *Salmonella* prevalence among pigs at farms and slaughterhouses, and in pork at the market place were 6%, 28%, and 29% in Northern Thailand, respectively [6]. A severe occurrence of *Salmonella*-contaminated pork in the same area was reported as 73.2% from markets [7]. These reports trigger a concern about how pathogen prevalence was increasing from farms to the markets. Additionally, there is a concern about how food-borne AMR pathogens could be transferred to humans [8], particularly how the AMR pathogens could be spread to dining tables. These concerns highlight the importance of procuring quantitative data of all steps in the development of food products, especially pork products.

The Hazard Analysis and Critical Control Point (HACCP) is a management system, which was introduced to ensure safe food products for consumers by controlling the biological, chemical, and physical hazards of meat, through monitoring and organizing the whole process. This management system was accepted after the United States Department of Agriculture’s (USDA, Washington, DC, USA) Food Safety and Inspection Service (FSIS, Maryland, USA) announced the rule in 1996 for chicken products [9]. Despite the great contribution this rule has made [10], many food-borne pathogens are still being reported [11,12]. It indicates that the HACCP system is still needed to help decrease the incidence of food-borne diseases. On the other hand, the non-HACCP level slaughterhouses would manufacture worse products than HACCP slaughterhouses, due to the lack of proper hygienic practices [13,14]. It seems these slaughter systems would lead to different prevalences, and the investigations of the two types of slaughterhouses would help in more useful information.

Moreover, the confusion about how AMR bacteria reach the dining table has continued for a long time. It has been believed that the AMR pathogens at the dining table are generated from the overuse of antimicrobials among farm animals [15,16]. Conversely, it has also been proposed that there is no direct connection of the AMR bacteria from the farm to the dining tables [17]. Many pathogens, such as *Salmonella* (non-typhoidal) cause self-limiting gastrointestinal disease, not usually present in/on muscle tissue. The food derived from farm animals is mostly associated with muscle tissue, and the exposure to the *Salmonella* should be limited. Unless of course, the bacteria from the intestines or the slaughterhouse environment contaminate the muscle tissue, which may happen during the slaughtering processes. Let us imagine three sources of contaminated products: (1) intrinsic infection due to the muscle tissue infected with pathogens; (2) direct contamination from the pig’s own gut flora; and (3) indirect contamination from the other pigs or the slaughterhouse environment. Of concern, are the most common contamination cases, which are from indirect contamination during the slaughtering process, because these indirect contaminations would be caused by a tiny error of practice (such as the incompleted disinfection). The indirect contaminations can be improved by completed disinfection at each step of the food development process. However, it is hard to ascertain the sources of contamination accurately, as the indirect contamination cases herein were assessed.

To evaluate the containments of AMR *Salmonella* contaminations of the two pig slaughterhouses, discovering whether the AMR *Salmonella*-contaminated products were directly related to farms, we instituted a paired sampling from two slaughterhouses (one HACCP and one non-HACCP). We report herein that both the HACCP and the non-HACCP slaughterhouses lead to a high level of indirect *Salmonella*-contaminated products.

## 2. Materials and Methods

Slaughtering methods: The slaughtering procedures of the two slaughterhouse varied. For HACCP slaughterhouse (HACCP SH), they included: stunning (shackling), sticking (bleeding), scalding, dehairing, gambrelling, polishing (singeing or shaving), pre-evisceration wash, bunging, head dropping, brisket opening, pull leaf fat, final trim, grade stand, and final wash. For the non-HACCP slaughterhouse (non-HACCP SH) procedures, they had not been normalized and they did not include a pre-evisceration wash nor a final wash step.

Pig sample collection and *Salmonella* isolation: For this study, 360 paired rectal and carcass samples were collected from 360 pigs at two slaughterhouses in Northeast Thailand from 2017 to 2018 (n = 720 total samples). Collection of these two samples from an individual pig represent the farm source *Salmonella* as determined by a positive rectal swab and contaminated product as determined by positive carcass swab of the pig.

One hundred and eighty paired samples were obtained from a single large-scale HACCP slaughterhouse (HACCP SH) that serviced six different farms (farms No. 1–6). The HACCP SH had good manufacturing practices (GMP) and HACCP certificates from the Department of Livestock Development (DLD, Bangkok, Thailand), Thailand. The other 180 paired samples were obtained from a small-scale slaughterhouse (non-HACCP SH) servicing twelve different farms (farms No. 7–18). The non-HACCP SH was registered with the local DLD office, Thailand. The HACCP SH was characterized by (1) restricting access to the slaughterhouse, (2) the assembly line, and 3) the cleaning and disinfection of critical points of the slaughtering process. The non-HACCP SH was characterized by (1) open access to the slaughterhouse, (2) disordered slaughtering processes, and 3) limited cleaning and disinfection steps. To determine whether the contaminated products in both slaughterhouses were produced indirectly from slaughtering, rectal samples were collected after stunning the pigs, and the carcass samples were collected by sterilized swabs covering a 600 cm^2^ area of each half pig carcass from the primal cuts after the final wash (from the HACCP SH) or from the primal cuts without a final wash (from the non-HACCP SH). All paired samples were meticulously matched and stored in an icebox for transportation.

*Salmonella* spp. isolation and detection were performed at the laboratory of Khon Kaen University Veterinary Teaching Hospital. The isolation of *Salmonella* spp. was performed, following the International Organization for Standardization (ISO, Geneva, Switzerland) standard procedure 6579-1:2002 [18]. The presumptive colonies (n = 3) of each sample were identified on XLD agar plates as *Salmonella* by their black color. Then the presumptive colonies were biochemically screened by use of triple sugar iron agar (TSI; Becton Dickinson, NJ, USA) and motility indole lysine medium (MIL; Becton Dickinson, NJ, USA) techniques. The identified *Salmonella* isolates were subsequently serotyped. The O (somatic) and H (flagellar) antigens of all rectum and carcass isolates were characterized by agglutination with hyperimmune sera in the World Health Organization (WHO) National Salmonella and Shigella Center for serotype identification, and the serotypes were assigned according to the Kauffmann–White scheme [19].

The slaughtering contamination rate (SCR) analysis for *Salmomella*: To evaluate the slaughterhouse efficiency at stopping *Salmonella* spread, the positive *Salmonella* rates of carcass and pig rectal isolates by farms were transformed into *Salmonella* slaughtering contamination rate (SCR). Then, the SCR means of two slaughterhouses were calculated.

The formula of the *Salmonella* SCR of one farm is:SCR=(contaminated product rate−infected animal rate)×100%.


The formula used to determine the contaminated product rate for a farm was determined to be: the number of *Salmonella* positive carcass samples divided by the total number of individuals collected, e.g., for carcass contamination rate of farm NO.1: 22 *Salmonella* positive samples divided by 30 collected individuals times 100% = 73.33%. The formula of the pig source rate for a farm was determined to be: the number of *Salmonella* positive rectum samples divided by the total number of individuals collected, e.g., for rectum (infected animal) *Salmonella* positive rate of farm NO.1: 29 positive samples divided by 30 times 100% = 96.67%. Then, the SCR of farm NO.1 is 73.33%–96.67% = −23.33%.

This formula reveals the difference of the *Salmonella*-contaminated product rate to *Salmonella* infected animal rate on one farm (Table 1). The SCR quantifies the efficiency at stopping *Salmonella* spread during the slaughtering process. A negative SCR means that the contaminated product rate is lower than the infected animal rate according to the formula, indicating that the contamination rate decreased. Conversely, a positive SCR indicates increasing contamination by the slaughtering process. The lower SCR illustrates a better slaughtering work at blocking pathogen spread.

Evaluation on *Salmonella* serotype diversities: The serotype diversities of rectal and carcass groups of the two slaughterhouses (including the serotypes and their amounts) were statistically analyzed by SPSS Statistics 17.0 (SPSS Inc, Chicago, IL, USA), using Pearson’s chi-squared test. *p* < 0.05 is considered significant.

Variation of antimicrobial resistance (AMR) patterns: All the *Salmonella* spp. isolates were analyzed for antimicrobial susceptibility against ampicillin (AMP) 10 μg, ciprofloxacin (CIP) 5 μg, chloramphenicol (CHL) 30 μg, nalidixic acid (NA) 30 μg, streptomycin (STR) 10 μg, sulfamethoxazole/trimethoprim (SXT) 25 μg, and tetracycline (TET) by disk diffusion test (BD Diagnostics, Sparks, Los Angeles, CA, USA), following the Clinical and Laboratory Standards Institute guidelines [20]. Two plates for each isolate were performed in the disk diffusion test. The significant differences of AMR resistance patterns between pig rectal and carcass groups in the two slaughterhouses were statistically analyzed by SPSS Statistics 17.0 (SPSS Inc, Chicago, IL, USA), using Pearson’s chi-squared test. *p* < 0.05 is considered significant.

The indirect contamination cases analysis: The sources of contaminations could be sorted into (1) intrinsic pathogen presence on the muscle tissue (the muscle tissue infected with pathogens); (2) direct contamination from the pig’s own gut flora; (3) indirect contamination from other pigs or the slaughterhouse environment. The intrinsic pathogen presence cases and the direct contamination cases would exhibit isolates common to both the rectal and carcass i.e., have the same *Salmonella* serotype and genetic type. The indirect contamination cases would consist of *Salmonella* spp. from the carcass sample that were not derived from the pig itself. Therefore, indirect contamination could be defined as the rest of the contamination cases except those of the same subtype pairs of isolates (assumed to be due to the intrinsic pathogen presence cases and direct contamination).

The same subtype pairs of isolates were attained following these procedures: (1) the serotypes of all pig rectal and carcass *Salmonella* isolates were screened for matched pairs of isolates by web-based software: calculate and draw custom Venn diagrams [21]; (2) the matched pair isolates were genetically confirmed by pulsed field gel electrophoresis (PFGE).

The PFGE analysis was performed following the CDC standard [22]. The DNA fragments were digested by *Xbal* and separated by a CHEF-DRIII pulsed-field electrophoresis system. The gels were stained with ethidium bromide and documented by ChemiDocTM XRS+ (Bio-Rad, Hercules, California, USA). The dendrogram was produced using band clustering with the dice coefficient similarity index of 1% optimization and 1% tolerance by the unweighted pair group method with arithmetic means (UPGMA) by BioNumerics version 7.6 (Applied Maths, Sint-Martens-Latem, Belgium).

## 3. Results and Discussion

The means of slaughtering contamination rate (SCR) of the two slaughterhouses: The paired samples from the HACCP SH consisted of 61.11% (110/180) and 12.78% (23/180) *Salmonella* positive rectal and carcass samples, respectively. Whereas, the paired samples from the non-HACCP SH consisted of 17.22% (31/180) and 36.67% (66/180) *Salmonella* positive rectal and carcass samples, respectively (Table 2). The SCR of *Salmonella* utilized here provides evidence on the efficiency at stopping the spread of *Salmonella* in the slaughterhouse. It is defined as the rate difference of *Salmonella*-contaminated products to *Salmonella* infected animals on one farm. The SCRs of each farm and the SCR means of two slaughterhouses were calculated (Table 1). The parameter “n” is the sample size of the investigated farms for one slaughterhouse. It influences the estimation power of the true SCR for the slaughterhouse. The SCR mean of HACCP SH was −48.89% ± 8.80% (n = 6); the SCR mean of non-HACCP SH was 14.31% ± 9.35% (n = 12), indicating this HACCP SH reduced the contamination level of *Salmonella*, while the non-HACCP SH contaminated more products. The SCR was utilized to evaluate the efficiency at stopping *Salmonella* contamination of meat products. We suggest it to be used as an important parameter assessed during the evaluation of contamination in slaughterhouses. In addition, the limited function of the HACCP system was exposed, that is some *Salmonella*-contaminated products were detected. Thus, additional processes are needed to decrease *Salmonella* contamination further. This study’s results are consistent with the other reports of positive *Salmonella* outcomes [11,12].

Antimicrobial resistance (AMR) of the two slaughterhouses: The susceptibility of *Salmonella* isolates to 7 antimicrobial agents was examined. The positive rectum source *Salmonella* spp. from the HACCP SH was resistant to AMP (76.36%), CHL (11.82%), STR (60.00%), SXT (17.27%), and TET (85.45%), while the isolates from carcass were resistant to AMP (95.65%), STR (95.65%), and TET (95.65%) (Table 3). From the non-HACCP SH, the pig rectum source were AMP (61.29%), CHL (3.23%), STR (29.03%), SXT (38.71%), and TET (64.52%) resistant, while the carcass source were AMP (69.70%), CHL (1.52%), STR (45.45%), SXT (15.15%), and TET (69.70%) resistant (Table 3). The most common resistance antimicrobials are similar to that reported in Thailand (AMP, STR, and TET resistance) [7,23]. The AMP, STR, and TET of the carcass were significantly higher than the rectum source in HACCP SH. The CHL and SXT of the carcass were significantly lower than the rectal group in HACCP SH. The STR of the carcass was significantly higher than the rectal group in non-HACCP SH. The SXT of the carcass was significantly lower than the rectal group in non-HACCP SH. It is unclear as to the antimicrobial resistance relationship in the two slaughterhouses and sample sources.

From the AMR pattern comparison result, the AMR patterns of the carcass group were significantly different from the pig rectal group (*p* < 0.01) in HACCP SH, while the AMR patterns of the carcass group were not significantly different from the pig rectal group (*p* > 0.05) in the non-HACCP SH. In the HACCP SH, the numbers of AMR patterns in carcass group (22 isolates with 1 pattern) were less than pig rectal group (96 isolates with 7 patterns), but the proportion of AMP-STR-TET in carcass group was higher than the pig rectal group. These results demonstrate that the AMR patterns in HACCP SH were reduced to one pattern after slaughtering. Nevertherless, the non-HACCP SH did not reduce the presence and diversity of AMR among the *Salmonella*-contaminated products, between the rectal group (22 isolates with 7 patterns) and carcass group (48 isolates with 8 patterns) (Table 4).

The indirect contamination cases of the two slaughterhouses: To assess indirect contamination cases (the rest results of contaminations of the same subtype pairs of isolates). All isolated serotypes were screened by web-based software: calculate and draw custom Venn diagrams based on serotyping and genetically confirmed by PFGE. The 12 rectal and 12 carcass isolates were serologically matched in pairs from the total 141 rectal and 89 carcass isolates. Ten of the 12 pairs were from HACCP SH, and two pairs were from non-HACCP SH (Figure 2). Then, the 12 serologically paired isolates were further assessed by PFGE (n = 24). The serotype matched pairs: R19 and C19, R341 and C341 were genetically different (Figure 3). The advanced indirect contamination cases were 14 from the total 23 contaminations in HACCP SH, and 65 from the total 66 contaminations in non-HACCP SH after PFGE confirmation. The rate of indirect contamination was 88.76% for the total contaminated carcasses (from two slaughterhouses) (79/89), revealing most cases were indirectly contaminated. Most of the indirect contaminations from HACCP slaughterhouse happened at one farm slaughtering, rated as 63.64% (14/22), total indirect contamination rate was 60.87% (14/23). Besides, almost all contaminations of each farm from the non-HACCP slaughtering were indirect contaminations, rated as 98.48% (65/66) (Figure 4). These results illustrate that both the non-HACCP SH and the HACCP SH maintain a high level of indirect contamination. These indirect contaminations could be prevented by the improvement of the slaughtering procedures. Therefore, this research proposed that the management of slaughtering procedures should be paid more attention. Some limitations to the research should be noted. (1) There could be some ‘same subtype pairs of isolates’ due to indirect contamination. This would lead to a lower estimation rate than the true indirect contamination rate. Despite the imprecise estimation of indirect contamination cases (the true indirect contamination rate is higher than the estimation), the current results still exposed a serious indirect contamination situation. (2) Although the isolation of *Salmonella* from each sample was repeated, there still might be some *Salmonella* spp. not discovered. Nevertheless, all data were analyzed based on the same level of the detection limit.

## 4. Conclusions

According to the results of this study, it is worth considering the slaughtering procedures used in controlling AMR *Salmonella* contamination of pig products from farms to the dining tables. Firstly, the HACCP slaughterhouse reduced the *Salmonella* occurrence, serotype numbers, and serotype diversity; whereas the non-HACCP slaughterhouse increased the *Salmonella* occurrence and introduced extra serotypes of *Salmonella* spp. into the meat products. Secondly, in both slaughterhouses, most of the AMR *Salmonella* spp.-contaminated products were associated with indirect contamination during slaughtering. This indicates that the AMR pathogens found at the dining tables may not directly relate to *Salmonella* from the farms. The results of this work may explain the phenomenon of the increasing *Salmonella* prevalence from farms to the markets [6,7]. These indirect contaminations may keep accumulating from slaughtering until selling. More work should focus on the individual evaluations of each slaughterhouse to improve their efficiencies at pathogen containment. Further, the food chain includes many steps from slaughtering to the final products. More studies on the other parts of the whole food chain process are needed to limit the risk of food contamination.

## Figures and Tables

**Figure 1 pathogens-09-00020-f001:**
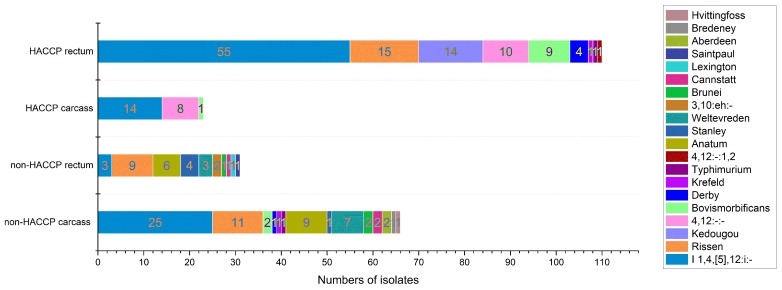
*Salmonella* spp. serotypes isolated from pig rectum and carcass between the two slaughterhouses. Note: The numbers of serotypes and the number of isolates from four different sources were different. The HACCP SH reduced the number of serotypes and the number of isolates from the HACCP rectum sample group (110 isolates with 9 serotypes) to carcass sample group (23 isolates with 3 serotypes), whereas the non-HACCP SH increased the number of serotypes and the number of isolates from the rectum sample group (31 isolates with 10 serotypes) to carcass sample group (66 isolates with 14 serotypes).

**Figure 2 pathogens-09-00020-f002:**
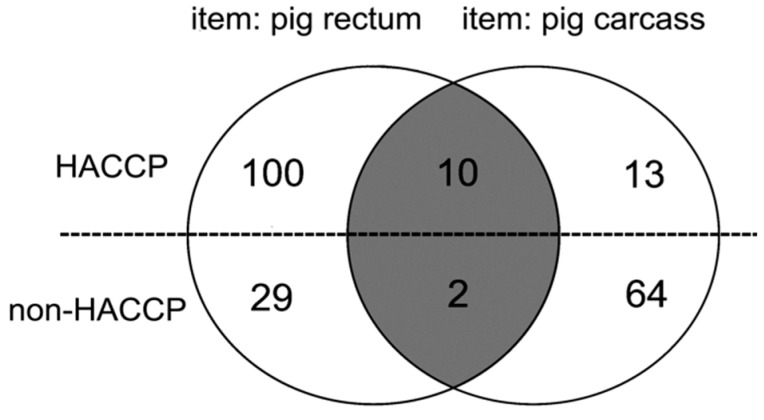
Serotype matched pairs of isolates between two slaughterhouses. Note: The 141 pig rectum and 89 carcass *Salmonella* isolates were screened by serotyping. Twelve paired isolates were matched, including ten pairs from HACCP SH and two pairs from non-HACCP SH. The serotype matched pair isolates need to be confirmed genetically. Only the serotype and genotype different carcass isolates were considered indirect contamination cases.

**Figure 3 pathogens-09-00020-f003:**
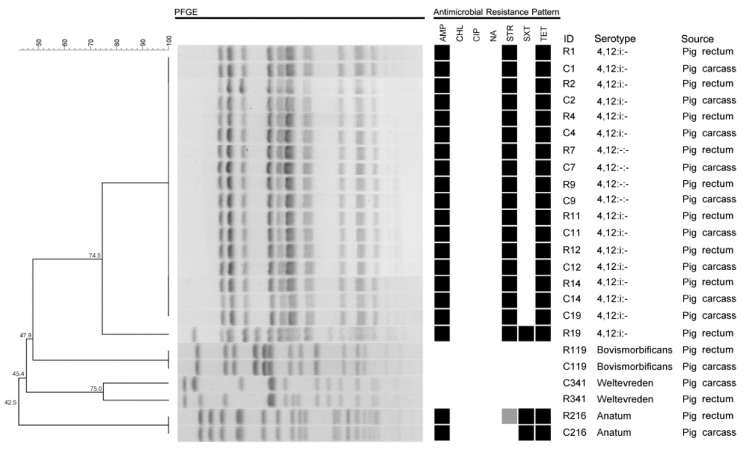
Genetic confirmation of the serotype matched pairs of isolates by PFGE (■ = resistance, ■ = intermediate). Note: (a) These non-matched pair isolates suggest that presence of indirect contamination cases. The isolates (R1, C1, R2, C2, R4, C4, R7, C7, R9, C9, R11, C11, R12, C12, R14, and C14) had a 100% similarity, indicating some isolates possibly were indirectly contaminated. It reveals a lower estimation rate than the true indirect contamination rate. (b) AMP: ampicillin, CIP: ciprofloxacin, CHL: chloramphenicol, NA: nalidixic acid, STR: streptomycin, SXT: sulfamethoxazole/trimethoprim, TET: tetracycline.

**Figure 4 pathogens-09-00020-f004:**
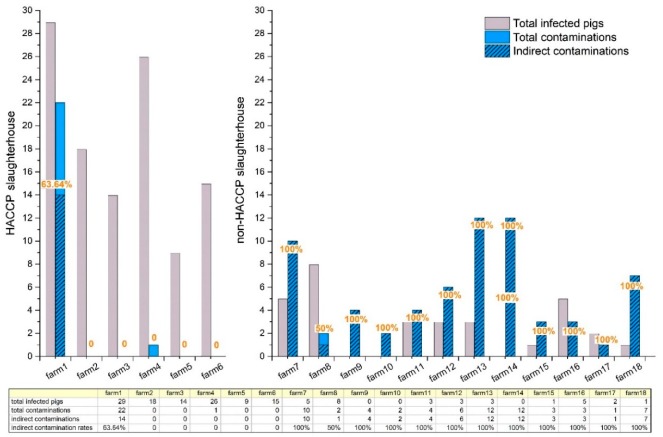
Indirect AMR *Salmonella* contamination rates of each farm. Note: The dashed area of the contaminations was shown as indirect contaminations. The indirect contaminated cases in HACCP SH happened only one time, while the indirect contaminated cases in non-HACCP SH were found every time. This reveals the HACCP SH is better than the non-HACCP SH in the containment of indirect contaminations.

**Table 1 pathogens-09-00020-t001:** Slaughtering contamination rate (SCRs) by farms between the two slaughterhouses.

	Source	Carcass	Rectum	SCR
HACCP SH	farm NO.1	73.33% (22/30)	96.67% (29/30)	−23.33%
	farm NO.2	0.00% (0/30)	60.00% (18/30)	−60.00%
	farm NO.3	0.00% (0/30)	46.67% (14/30)	−46.67%
	farm NO.4	3.33% (1/30)	86.67% (26/30)	−83.33%
	farm NO.5	0.00% (0/30)	30.00% (9/30)	−30.00%
	farm NO.6	0.00% (0/30)	50.00% (15/30)	−50.00%
	Mean ± standard error of mean		−48.89% ± 8.80%	
non-HACCP SH	farm NO.7	33.33% (10/30)	16.67% (5/30)	16.67%
	farm NO.8	20.00% (2/10)	80.00% (8/10)	−60.00%
	farm NO.9	40.00% (4/10)	0.00% (0/10)	40.00%
	farm NO.10	20.00% (2/10)	0.00% (0/10)	20.00%
	farm NO.11	40.00% (4/10)	30.00% (3/10)	10.00%
	farm NO.12	60.00% (6/10)	30.00% (3/10)	30.00%
	farm NO.13	60.00% (12/20)	15.00% (3/20)	45.00%
	farm NO.14	60.00% (12/20)	0.00% (0/20)	60.00%
	farm NO.15	15.00% (3/20)	5.00% (1/20)	10.00%
	farm NO.16	30.00% (3/10)	50.00% (5/10)	−20.00%
	farm NO.17	10.00% (1/10)	20.00% (2/10)	−10.00%
	farm NO.18	35.00% (7/20)	5.00% (1/20)	30.00%
	Mean ± standard error of mean		14.31% ± 9.35%	

Note: (a) The formula of SCR = (contaminated product rate − infected animal rate) × 100% reveals the value difference of the *Salmonella*-contaminated product rate to the *Salmonella* infected animal rate of each farm. ‘Rectum’ stands for the *Salmonella* infected animal rate. ‘Carcass’ means the *Salmonella*-contaminated product rate. (b) Interestingly, although the rising tendency of contaminations in non-HACCP SH was presented, three reduced contamination cases were observed. These results suggest that a single SCR analysis of one-time slaughtering is inadequate to estimate the true SCR for the slaughterhouse. Serotype diversity of the two slaughterhouses: From the serotype diversity result, the serotype diversity including the serotypes and their amounts of carcass group (23 isolates with 3 serotypes) was significantly lower than the pig rectal sample group (110 isolates with 9 serotypes) (*p* < 0.05) in HACCP SH; the diversity of the carcass group (66 isolates with 14 serotypes) was not significantly different from the pig rectal group (31 isolates with 10 serotypes) (*p* > 0.05) in the non-HACCP SH. In particular, six serotypes were prevented from entering the food chain by HACCP slaughtering process. In non-HACCP SH, seven extra serotypes were introduced into the food chain, and three serotypes were blocked. The HACCP SH procedures can reduce the serotype diversities from entering into the food chain, but the non-HACCP SH introduced extra serotypes into the food chain (Table 2 and Figure 1).

**Table 2 pathogens-09-00020-t002:** *Salmonella* spp. serotypes from two slaughterhouses.

Slaughterhouse	Source	No. of Isolates	Serotypes (n)
HACCP SH	Rectum	61.11% (110/180)	Enterica ser. 1,4,[5],12:i:- (55), Rissen (15), Kedougou (14), Enterica ser. 4,12:-:- (10), Bovismorbificans (9), Derby (4), Krefeld (1), Typhimurium (1), Enterica ser. 4,12:-:1,2 (1)
non-HACCP SH	Rectum	17.22% (31/180)	Rissen (9), Anatum (6), Stanley (4), Enterica ser. 1,4,[5],12:i:- (3), Weltevreden (3), Enterica 3,10:eh:- (2), Brunei (1), Cannstatt (1), Lexington (1), Saintpaul (1)
HACCP SH	Carcass	12.78% (23/180)	Enterica ser. 1,4,[5],12:i:- (14), Enterica ser. 4,12:-:- (8), Bovismorbificans (1)
non-HACCP SH	Carcass	36.67% (66/180)	Enterica ser. 1,4,12:i:- (25), Rissen (11), Anatum (9), Weltevreden (7), Aberdeen (2), Bovismorbificans (2), Brunei (2), Cannstatt (2), Bredeney (1), Derby (1), Hvittingfoss (1), Krefeld (1), Stanley (1), Typhimurium (1)
Total samples		31.94% (230/720)	

Note: These results show that among pigs from the 6 farms that were provided to the HACCP slaughterhouse (HACCP SH), high carrier status of *Salmonella* (61.11%) was found. However, in the HACCP slaughterhouse, only 12.78% of the carcass samples were *Salmonella* positive, suggesting a reduced contamination level. In contradiction to these observations of the HACCP SH, the presence of *Salmonella* found among pigs from the 12 farms provided to non-HACCP slaughterhouse (non-HACCP SH) was only 17.22 %, but the contamination proportion was increased to 36.67%.

**Table 3 pathogens-09-00020-t003:** Antimicrobial resistance agents of *Salmonella* spp. from two slaughterhouses.

Groups	Serotype (n)	Antimicrobial Resistance Agents (%)						
		AMP	CHL	CIP	NA	STR	SXT	TET
Rectum of HACCP	I 1,4,[5],12:i:- (55) Rissen (15) Kedougou (14) 4,12:-:- (10) Bovismorbificans (9) Derby (4) Krefeld (1) Typhimurium (1) 4,5,12:-:1,2 (1)	51 (60.71) 15 (17.86) 1 (1.19) 10 (11.90) 0 4 (4.76) 1 (1.19) 1 (1.19) 1 (1.19)	0 0 13 (100) 0 0 0 0 0 0	0 0 0 0 0 0 0 0 0	0 0 0 0 0 0 0 0 0	50 (75.76) 4 (6.06) 0 9 (13.64) 0 0 1 (1.52) 1 (1.52) 1 (1.52)	2 (10.53) 12 (63.16) 0 1 (5.26) 0 4 (21.05) 0 0 0	52 (55.32) 15 (15.96) 14 (14.89) 10 (10.64) 0 0 1 (1.06) 1 (1.06) 1 (1.06)
	Total (110)	84 (76.36)	13 (11.82)	0	0	66 (60.00)	19 (17.27)	94 (85.45)
Carcass of HACCP	4,12:i:- (14) 4,12:-:- (8) Bovismorbificans (1)	14 (63.64) 8 (36.36) 0	0 0 0	0 0 0	0 0 0	14 (63.64) 8 (36.36) 0	0 0 0	14 (63.64) 8 (36.36) 0
	Total (23)	23 (95.65)	0	0	0	23 (95.65)	0	23 (95.65)
Rectum of non-HACCP	Rissen (9) Anatum (6) Stanley (4) I 1,4,[5],12:i:- (3) Weltevreden (3) 3,10:eh:- (2) Brunei (1) Cannstatt (1) Lexington (1) Saintpaul (1)	9 (29.03) 6 (19.35) 1 (3.23) 3 (9.68) 0 0 0 0 0 0	0 0 0 0 0 0 0 0 0 0	0 0 0 0 0 0 0 0 0 0	0 0 1 (3.23) 0 0 0 0 0 0 0	2 (6.45) 1 (3.23) 2 (6.45) 3 0 0 0 0 1 (3.23) 0	6 (19.35) 6 0 0 0 0 0 0 0 0	9 (29.03) 6 (19.35) 1 (3.23) 3 (9.68) 0 1 (3.23) 0 0 0 0
	Total (31)	19 (61.29)	0	0	1 (3.23)	9 (29.03)	12 (38.71)	20 (64.52)
Carcass of non-HACCP	I 1,4,[5],12:i:- (25) Rissen (11) Anatum (9) Weltevreden (7) Aberdeen (2) Bovismorbificans (2) Brunei (2) Cannstatt (2) Bredeney (1) Derby (1) Hvittingfoss (1) Krefeld (1) Stanley (1) Typhimurium (1)	24 (36.36) 11 (16.67) 8 (12.12) 0 0 0 0 0 0 1 (1.52) 0 1 (1.52) 0 1 (1.52)	0 0 0 0 0 0 0 0 0 1 (1.52) 0 0 0 0	0 0 0 0 0 0 0 0 0 0 0 0 0 0	0 0 0 0 0 0 0 0 0 0 0 0 0 0	24 (36.36) 3 (4.55) 0 0 0 0 1 (1.52) 0 0 0 0 1 (1.52) 0 1 (1.52)	1 (1.52) 2 (3.03) 6 (9.09) 0 0 0 0 0 0 1 (1.52) 0 0 0 0	23 (34.85) 11 (16.67) 9 (13.64) 0 0 0 0 0 0 1 (1.52) 0 1 (1.52) 0 1 (1.52)
	Total (66)	46 (69.70)	1 (1.52)	0	0	30 (45.45)	10 (15.15)	46 (69.70)

**Table 4 pathogens-09-00020-t004:** Antimicrobial resistance (AMR) patterns of *Salmonella* spp. from the two slaughterhouses.

Pattern	HACCP (%) (n/N)		non-HACCP (%) (n/N)		Total (%) (n/N)
	Rectum	Carcass	Rectum	Carcass	
AMP-STR-TET	57.27 (63/110)	95.65 (22/23)	19.35 (6/31)	40.91 (27/66)	51.30 (118/230)
AMP-SXT-TET	11.82 (13/110)	0	35.48 (11/31)	12.12 (8/66)	13.91 (32/230)
CHL-TET	10.91 (12/110)	0	0	0	5.22 (12/230)
AMP-TET	0	0	3.23 (1/31)	12.12 (8/66)	39.13 (9/230)
AMP-STR-SXT-TET	1.82 (2/110)	0	3.23 (1/31)	1.52 (1/66)	1.74 (4/230)
AMP-SXT	3.64 (4/110)	0	0	0	1.74 (4/230)
AMP-STR	0.91 (1/110)	0	0	1.52 (1/66)	0.86 (2/230)
TET	0	0	3.23 (1/31)	1.52 (1/66)	0.86 (2/230)
STR	0	0	3.23 (1/31)	1.52 (1/66)	0.86 (2/230)
AMP-CHL-SXT-TET	0	0	0	1.52 (1/66)	0.43 (1/230)
AMP-CHL-TET	0.91 (1/110)	0	0	0	0.43 (1/230)
NA-STR	0	0	3.23 (1/31)	0	0.43 (1/230)
Total (%) (n/N)	87.27 (96/110)	95.65 (22/23)	70.96 (22/31)	72.73 (48/66)	

Note: Although the positive AMR patterns in HACCP SH were reduced from seven patterns to one (AMP-STR-TET), this also increased the prevalence of the AMR to AMP, STR, TET.

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
