# Peer review of "Evaluation of the Containment of Antimicrobial-Resistant Salmonella Species from a Hazard Analysis and Critical Control Point (HACCP) and a Non-HACCP Pig Slaughterhouses in Northeast Thailand"

_pathogens, 2019, doi:10.3390/pathogens9010020_

Round 1

Reviewer 1 Report

In the manuscript „Evaluation of the Containment of Antimicrobial-Resistant Salmonella species of a Hazard Analysis and Critical Control Point (HACCP) and a Non-HACCP Pig Slaughterhouses in Northeast Thailand” of Xin Wu and colleagues, the authors reported on a phenotypic and genotypic comparison of Salmonella isolates from a HACCP slaughterhouse (HACCP-SH) and a non-HACCP slaughterhouse (non-HACCP-SH) in Northeast Thailand. The results were interpreted to show that both the slaughterhouses keep a high level of indirect contamination, while the HACCP-SH decreases Salmonella contaminations and reduces the AMR patterns, the non-HACCP-SH increases contaminations.

This reviewer feels that the authors presented an interesting study on the differences of isolates of HACCP and non-HACCP slaughterhouses. The manuscript provides a good overview and is well written.

Line 121 to 126: Please include the main differences of both slaughterhouses… are there any differences in the processes, temperature, other conditions…

Line 129: This reviewer feel that the majority of this section could be omitted only by citing meaningful references…

Author Response

Dear Reviewers and the Assistant Editor Nancy Ma,

We are grateful for your helpful comments on our manuscript. We have tried our best to modify the manuscript according to the two reviewers’ comments. We have addressed all the issues raised and have modified the paper accordingly. Below is a summary of the changes we performed and our responses to the reviewers’ comments and recommendations.

Sincerely,

Xin Wu

According to the reviewer A:

Point 1: Line 121 to 126: Please include the main differences of both slaughterhouses… are there any differences in the processes, temperature, other conditions…

Response 1: At the line 121 to 126 (previous manuscript), we have added the sentence saying that “The HACCP SH was characterized by 1) restricting access to the slaughterhouse, 2) the assembly line, and 3) the cleaning and disinfection of critical points of the slaughtering process. The non-HACCP SH was characterized by 1) open access to the slaughterhouse, 2) disordered slaughtering processes, and 3) limited cleaning and disinfection steps.

Point 2: Line 129: This reviewer feel that the majority of this section could be omitted only by citing meaningful references…

Response 2: At the line 129 (previous manuscript), we have omitted the description of standard isolation method, only cited the reference.

Reviewer 2 Report

Review of the article: Evaluation of the Containment of Antimicrobial-Resistant Salmonella species of a Hazard Analysis and Critical Control Point (HACCP) and a Non-HACCP Pig Slaughterhouses in Northeast Thailand

Manuscript ID: pathogens-657859

The proposed idea of the research is very interesting and not very common in the literature. I think that the manuscript could be interesting for many researchers who are interested in food microbiology. Some of the results are really interesting. However, some important changes in the text of the manuscript would be required. In my opinion some fragments are not clear. I also do not agree with some interpretations presented by the authors. Below I have presented detailed comments.   

Detailed comments

Abstract – I was confused reading abstract, e.g. SCR is not explained by the authors, so the subsequently presented results are not clear – I was able to understand it after reading the whole text of the manuscript. Next element which is not clear, the authors have written: “the HACCP-SH decreased contamination (SCR: -48.89% ± 8.80%, n=6)”. But they did not explain meaning of “n” – the same problem I found in description of results. In my opinion it is not well presented it should be clearly what the “n” means – or it should be omitted at all. It also should be clearly written what is rectal group – contamination/resistance of strains isolated directly from animals (from the farm) and what is carcass group. It is explained in the text, but not in the abstract, so it is not clear.

Introduction – generally well written and very interesting (lines 62-65). I only do not agree that HACCP is dedicated only for meat industry, it is more general system used in production of food products.

Materials and methods

I do not have important critical remarks about methodology, however, in the part of manuscript where the disc diffusion assay is presented (lines 165-173) it should be written that these are amounts of antibiotics in the discs.

Results – lines 207-211 – above I have written about presenting parameter “n” – please consider if it is necessary, or it should be explained.

Table 2, lines 216-217 – I would like to notice that in two of cases (non-HACCP slaughterhouse) important reduction of contamination was observed; it should be mentioned and commented.

Titles/descriptions of all Tables and Figures should be more detailed and more informative – general comment

Table 3 – it is written “Antimicrobial resistance agents” – it should be changed

Some more comments would be required in the case of figures 3 and 4.

Conclusions – acceptable, however, this part is long and some information could be moved to the section Results and discussion. I also think that authors could use more citations for comparison of their results with the outcomes of investigations presented by other authors.

Final opinion – acceptance after major revision

Author Response

Dear Reviewers and the Assistant Editor Nancy Ma,

We are grateful for your helpful comments on our manuscript. We have tried our best to modify the manuscript according to the two reviewers’ comments. We have addressed all the issues raised and have modified the paper accordingly. Below is a summary of the changes we performed and our responses to the reviewers’ comments and recommendations.

Sincerely,

Xin Wu

According to the reviewer B:

Point 1: Abstract – I was confused reading abstract, e.g. SCR is not explained by the authors, so the subsequently presented results are not clear – I was able to understand it after reading the whole text of the manuscript. Next element which is not clear, the authors have written: “the HACCP-SH decreased contamination (SCR: -48.89% ± 8.80%, n=6)”. But they did not explain meaning of “n” – the same problem I found in description of results. In my opinion it is not well presented it should be clearly what the “n” means – or it should be omitted at all. It also should be clearly written what is rectal group – contamination/resistance of strains isolated directly from animals (from the farm) and what is carcass group. It is explained in the text, but not in the abstract, so it is not clear.

Response 1: Abstract part:

We have explained:

1) SCR (to evaluate the contamination level by averaged farm SCRs);

2) Rectal group (representing the farm pig status);

3) Carcass group (representing the contamination).

The parameter “n” has been explained in the text.

Point 2: Introduction – generally well written and very interesting (lines 62-65). I only do not agree that HACCP is dedicated only for meat industry, it is more general system used in production of food products.

Response 2: Introduction part:

We have changed the words from “meat products” to “food products

Point 3: Materials and methods – I do not have important critical remarks about methodology, however, in the part of manuscript where the disc diffusion assay is presented (lines 165-173) it should be written that these are amounts of antibiotics in the discs.

Response 3: Materials and methods part:

We have added the sentence saying that “Two plates for each isolate were performed in the disk diffusion test.

Point 4: Results – lines 207-211 – above I have written about presenting parameter “n” – please consider if it is necessary, or it should be explained.

Response 4: Results part:

At lines 207 to 211 (previous manuscript), we have explained the parameter “n” saying that “The parameter “n” is the sample size of the investigated farms for one slaughterhouse. It influences the estimation power of the true SCR for the slaughterhouse.

Point 5: Table 2, lines 216-217 – I would like to notice that in two of cases (non-HACCP slaughterhouse) important reduction of contamination was observed; it should be mentioned and commented.

Response 5: At Table 2, lines 216-217 (previous manuscript), we have mentioned the decreased contamination rate in the non-HACCP slaughterhouse, and saying that “Interestingly, although the rising tendency of contaminations in non-HACCP SH was presented, three reduced contamination cases were observed. These results suggest that a single SCR analysis of one-time slaughtering is inadequate to estimate the true SCR for the slaughterhouse.

Point 6: Titles/descriptions of all Tables and Figures should be more detailed and more informative – general comment. Some more comments would be required in the case of figures 3 and 4.

Response 6: According to the suggestions “Titles/descriptions of all Tables and Figures should be more detailed and more informative – general comment.”, and “Some more comments would be required in the case of figures 3 and 4.”

We have added more information or comments:

For Table 1 added “Note: These results show that among pigs from the 6 farms that were provided to the HACCP SH, a high carrier status of Salmonella (61.11%) was found. However, in the HACCP slaughterhouse, only 12.78% of the carcass samples were Salmonella positive, suggesting a reduced contamination level. In contradiction to these observations of the HACCP SH, the presence of Salmonella found among pigs from the 12 farms provided to non-HACCP SH was only 17.22 %, but the contamination proportion was increased to 36.67%. Table 4 added “Note: Although the positive AMR patterns in HACCP SH were reduced from seven patterns to one (AMP-STR-TET), this also increased the prevalence of the AMR to AMP, STR, TET. Figure 1 added “Note: The numbers of serotypes and the number of isolates from four different sources were different. The HACCP SH reduced the number of serotypes and the number of isolates from the HACCP rectum sample group (110 isolates with 9 serotypes) to carcass sample group (23 isolates with 3 serotypes), whereas the non-HACCP SH increased the number of serotypes and the number of isolates from rectum sample group (31 isolates with 10 serotypes) to carcass sample group (66 isolates with 14 serotypes). Figure 2 added “Note: The 141 pig rectum and 89 carcass Salmonella isolates were screened by serotyping. Twelve paired isolates were matched, including ten pairs from HACCP SH and two pairs from non-HACCP SH. The serotype matched pair isolates need be confirmed genetically. Only the serotype and genotype different carcass isolates were considered indirect contamination cases. Figure 3 added “Note: a. These non-matched pair isolates suggest that presence of indirect contamination cases. The isolates (R1, C1, R2, C2, R4, C4, R7, C7, R9, C9, R11, C11, R12, C12, R14, & C14) had a 100% similarity, indicating some isolates possibly were indirectly contaminated. It reveals a lower estimation rate than the true indirect contamination rate. b. AMP: ampicillin, CIP: ciprofloxacin, CHL: chloramphenicol, NA: nalidixic acid, STR: streptomycin, SXT: sulfamethoxazole/trimethoprim, TET: tetracycline. Figure 4 added “Note: The dashed area of the contaminations was shown as indirect contaminations. The indirect contaminated cases in HACCP SH happened only one time, while the indirect contaminated cases in non-HACCP SH were found every time. This reveals the HACCP SH is better than the non-HACCP SH in the containment of indirect contaminations.

Point 7: Table 3 – it is written “Antimicrobial resistance agents” – it should be changed

Response 7: The title of Table 3 has been changed to “Antimicrobial resistance agents of Salmonella spp. from two slaughterhouses.

Point 8: Conclusions – acceptable, however, this part is long and some information could be moved to the section Results and discussion. I also think that authors could use more citations for comparison of their results with the outcomes of investigations presented by other authors.

Response 8: Conclusions part:

We have move the sentence “the limited function of the HACCP system was exposed that is some Salmonella contaminated products were detected. Thus, additional processes are needed to decrease Salmonella contamination further. This study’s results are consistent with the other reports of positive Salmonella outcomes [11, 12].” to the result & discussion part, at the end of the section “The means of slaughtering contamination rate (SCR) of the two slaughterhouses”. Added the comparison with other outcomes saying that “The results of this work may explain the phenomenon of the increasing Salmonella prevalence from farms to the markets [6, 7]. These indirect contaminations may keep accumulating from slaughtering until selling. Moved the sentence “These indirect contaminations could be prevented by the improvement of the slaughtering procedures. Therefore, this research proposed that the management of slaughtering procedures should be paid more attention to by food product leadership.” to the result & discussion part, at the end of the section “The indirect contamination cases of the two slaughterhouses” Omitted the sentence “Besides, the policies for the whole food chain should provide better procedures for management to stop AMR pathogens spreading.

Round 2

Reviewer 2 Report

I am fully satisfied with the responses presented by the authors of the manuscript. The manuscript can be accepted.

This manuscript is a resubmission of an earlier submission. The following is a list of the peer review reports and author responses from that submission.

Round 1

Reviewer 1 Report

In the manuscript „ Evaluation of the Containment of Antimicrobial-Resistant Salmonella species by a Hazard Analysis and Critical Control Point (HACCP) System and a Non-HACCP System utilizing Pig Slaughterhouses in Northeast Thailand” of Xin Wua and colleagues, the authors reported on microbiological and molecular investigations for comparison slaughterhouses using HACCP and non-HACCP systems. Based on the results of the study, the authors found that both systems keep a high level of indirect contamination, while the HACCP-SH decreases Salmonella contaminations and reduces the AMR patterns, the non-HACCP-SH increases contaminations.

This reviewer feels that the manuscript provide here represents a very good study. It is the opinion of this reviewer that the manuscript is ready for publication after minor changes. I would like to recommend providing figure 2 with colors instead of shadings. Furthermore, the used abbreviation in the manuscript needs to be explained…

In figure 4 the abbreviation for the antimicrobials are unclear. Furthermore the quality of the image needs to be improved.

Nevertheless, a very nice story!

Reviewer 2 Report

This article was focused on the occurrence of AMR Salmonella species from pig, especially comparing between HACCP and non HACCP slaughterhouses. Enough samples from 18 farms were collected and analyzed. However, the effect of HACCP implemented on AMR prevalence could not determined the significance since only one slaughterhouse was compared. To provide scientific significances on the merit of HACCP system, you should have enough number of slaughterhouses in addition to the pig samples. 

Reviewer 3 Report

Minor comments

Line 47-48. A severe occurrence of pork in the same area was; should read “A severe occurrence of salmonella contaminated pork in the same area was”.

Line 54: change “produce” to ‘’ensure’’

Line 54: change “produce” to ‘’ensure’’

Line 55: change “though” to ‘’through’’

Line 56: give full meaning of USDA’s and the abbreviation in brackets

Line 63: “the confusion about how sources of AMR bacteria reach” should be changed to “the confusion about how AMR bacteria reach”

Line 89: give full meaning of GMP

Line 105: “onto selected by Xylose Lysine Deoxycholate agar” should read “ onto Xylose Lysine Deoxycholate agar”

Line 89: give full meaning of ISO

Line 141: “ indirect contamination from other pig individuals” should be changed to  “ indirect contamination from other pig(s)”

Line 154: give full meaning of CDC

Major concern

Line 79-80: a study to assess one slaughterhouse that utilizes HACCP and one that did not.

Analyzing data based on the practices of just 1 HACCP house and 1 none HACCP house will not give an accurate effect of HACCP and none HACCP on curtailing contamination. This research was not properly envisage and designed.This study is very limiting and results obtained can be very misleading.